# Upfront Surgery versus Neoadjuvant Perioperative Chemotherapy for Resectable Colorectal Liver Metastases: A Machine-Learning Decision Tree to Identify the Best Potential Candidates under a Parenchyma-Sparing Policy

**DOI:** 10.3390/cancers15030613

**Published:** 2023-01-18

**Authors:** Simone Famularo, Flavio Milana, Matteo Cimino, Eloisa Franchi, Mario Giuffrida, Guido Costa, Fabio Procopio, Matteo Donadon, Guido Torzilli

**Affiliations:** 1Department of Biomedical Sciences, Humanitas University, 20072 Pieve Emanuele, Italy; 2Division of Hepatobiliary Surgery, Department of Surgery, IRCCS Humanitas Research Hospital, 20089 Rozzano, Italy; 3Surgical Data Science Team, Institut de Recherche sur les Cancers de l’Appareil Digestif (IRCAD), 67000 Strasbourg, France; 4General Surgery Unit, Department of Medicine and Surgery, University of Parma, 43121 Parma, Italy; 5Department of Health Sciences, Università del Piemonte Orientale, 28100 Novara, Italy; 6Department of General Surgery, University Maggiore Hospital Della Carità, 28100 Novara, Italy

**Keywords:** liver surgery, colorectal liver metastases, upfront surgery, neoadjuvant chemotherapy, machine-learning

## Abstract

**Simple Summary:**

For patients with colorectal liver metastases (CLM), it is doubtful which treatment could be better between neoadjuvant chemotherapy followed by liver resection (NEOS) and upfront surgery (UPS). Our aim was to select the candidates who may benefit more from one or another treatment developing a machine-learning model. To do so, 448 patients were analyzed, and baseline differences were levelled out thanks to an inverse probability weighting analysis. Survival rates and risk factors were estimated for the two generated pseudo-populations. The best potential treatment (BPT) for each patient was determined thanks to a mortality risk model built by Random-Forest. BPT-upfront and BPT-neoadjuvant candidates were automatically selected with the development of a classification –and –regression tree (CART). At CART, planning R1vasc surgery, primitive tumor localization, number of metastases, sex, and pre-operative CEA were the factors addressing the candidates to BPT. Thanks to the decision tree algorithm, patients may be automatically assigned to the BPT based on their tailored risk of mortality.

**Abstract:**

Addressing patients to neoadjuvant systemic chemotherapy followed by surgery rather than surgical resection upfront is controversial in the case of resectable colorectal –liver metastases (CLM). The aim of this study was to develop a machine-learning model to identify the best potential candidates for upfront surgery (UPS) versus neoadjuvant perioperative chemotherapy followed by surgery (NEOS). Patients at first liver resection for CLM were consecutively enrolled and collected into two groups, regardless of whether they had UPS or NEOS. An inverse –probability weighting (IPW) was performed to weight baseline differences; survival analyses; and risk predictions were estimated. A mortality risk model was built by Random-Forest (RF) to assess the best –potential treatment (BPT) for each patient. The characteristics of BPT-upfront and BPT-neoadjuvant candidates were automatically identified after developing a classification –and –regression tree (CART). A total of 448 patients were enrolled between 2008 and 2020: 95 UPS and 353 NEOS. After IPW, two balanced pseudo-populations were obtained: UPS = 432 and NEOS = 440. Neoadjuvant therapy did not significantly affect the risk of mortality (HR 1.44, 95% CI: 0.95–2.17, *p* = 0.07). A mortality prediction model was fitted by RF. The BPT was NEOS for 364 patients and UPS for 84. At CART, planning R1vasc surgery was the main factor determining the best candidates for NEOS and UPS, followed by primitive tumor localization, number of metastases, sex, and pre-operative CEA. Based on these results, a decision three was developed. The proposed treatment algorithm allows for better allocation according to the patient’s tailored risk of mortality.

## 1. Introduction

Surgical resection represents the only curative treatment for colorectal liver metastases (CLMs), with overall survival (OS) rates at 5 and 10 years of about 50% and 35%, respectively [1]. The latest ESMO (European Society of Medical Oncology) consensus guidelines [2] allow upfront surgery for patients with “technically easy” and “excellent oncologic prognosis” CLMs. It has been reported that only a minority of CLMs are upfront resectable for technical reasons [3,4]. In such a scenario, conversion chemotherapy protocols were considered a fundamental tool in the therapeutic pathway of the disease [5]. However, the concept of technical feasibility has drastically changed in the last ten years with the introduction of new concepts of parenchyma-sparing surgery, making resection suitable even in cases of complex liver involvement [6]. Under this surgical policy, a fine intraoperative ultrasound guide combined with the recent evidence regarding the optimal outcome in cases of tumor vessel detachment (R1 vascular resection) opened the door to new surgical solutions, safely expanding the resectability [7]. In cases of feasible surgery, perioperative neoadjuvant chemotherapy has been recommended as the “most reasonable” approach when the oncologic prognosis is not sharply clear, or the risk of recurrence after surgery is increased [2]. However, no significant differences at 3- and 5-year OS have been demonstrated [8] in such cases, chemotherapy-induced liver toxicity and chemotherapy failure with tumor progression may exclude patients from surgery [9,10]. Anyway, no practical criteria have been established regarding which patients may benefit more from neoadjuvant protocols, and whether to deem an upfront resectable patient to a preoperative protocol is still a point of debate. The aim of this study was to report a single-center experience for better clarification of what could be the best potential treatment for such complex cases through the production of a best treatment allocation model based on a machine-learning approach.

## 2. Materials and Methods

### 2.1. Study Overview, Patient Selection, and Study Design

The present is a retrospective study based on a prospectively collected institutional dataset from Humanitas Research Hospital IRCCS (Rozzano, Milan, Italy). Results are reported according to the principles of Strengthening the Reporting of Observational Studies in Epidemiology (STROBE) [11]. All consecutive adult patients (age ≥18 years) treated for liver metastases derived from primary colorectal cancer (CLM) from 2008 to 2020 were considered for enrolment. At our centre, all cases are routinely discussed in a multidisciplinary setting, including liver surgeons, oncologists, dedicated radiologists, hepatologists, and radiotherapists [12]. The indications among UPS and NEOS were considered specifically for every single patient, as were the sum of the patients’ underlying condition, oncologic and medical history, and local protocols. Inclusion criteria of this study were: (1) first radiological diagnosis of CLM treated for the first time with liver resection, (2) a feasible liver resection under a parenchyma sparing approach; and exclusion criteria were: (1) missing data on the follow-up variables, (2) a progression-disease after neoadjuvant chemotherapy discovered at the radiological imaging (either CT or MRI scans) according to the RECIST [13] criteria, and (3) classic major hepatectomies as in case of right or extended right hepatectomy, left or extended left hepatectomy or trisectionectomy.

First, our study compared the overall survival among patients submitted to neoadjuvant chemotherapy followed by liver resection versus upfront liver resection after weighing the oncologic risk among groups. Second, a machine-learning approach was employed to develop a mortality prediction model to define the best potential treatment for each patient according to his/her oncologic and clinical characteristics, and then a decision tree was developed to select the most effective variables and to highlight who could benefit the most from one treatment rather than the other in the era of parenchyma-sparing vessel-guided surgery.

### 2.2. Definitions and Follow-Up Protocol

In our centre, hepatic resection is performed according to parenchyma-sparing vessel-guided surgery [14] relying on R1vascular [7], intraoperative ultrasound-guided navigation [15], and communicating veins [16]. Liver resection was scheduled four to six weeks after the end of chemotherapy (six weeks in patients receiving anti-VEGF targeted therapies). Biochemical values were obtained within two weeks of the assigned treatment. The number and size of nodules were assessed preoperatively by magnetic resonance imaging (MRI), preferentially with a hepatospecific contrast agent, and examined by an expert and dedicated radiologist; multiphase contrast computed tomography (CT) was performed as an alternative when MRI could not be performed. Postoperative complications were recorded using the Clavien–Dindo classification [17]. Patients treated with chemotherapy were submitted to the standard protocol as follows: oxaliplatin, irinotecan, and irinotecan plus oxaliplatin-based chemotherapy. Targeted chemotherapy was added according to RAS mutation status and evaluated on the basis of a case-by-case multidisciplinary discussion. All patients received at least two cycles of chemotherapy and were evaluated with an MRI or CT scan every 4–6 cycles.

The tumor response to neoadjuvant chemotherapy was evaluated according to the RECIST criteria [18].

All patients were followed up using local protocols, which included measurement of serum tumor markers (Ca19.9 and CEA), abdominal ultrasound, CT or MRI, and outpatient visits. OS was defined as the time from the date of the assigned treatment to any cause of death. Patient surveillance was closed at the end of February 2021.

### 2.3. Statistical Analysis

Normal distribution was tested by the Kolmogorov–Smirnov test. Data were presented as frequency and percentage in the case of categorical variables or by median and interquartile range in the case of continuous variables. Mann–Whitney and Fisher’s tests were used to compare baseline patient characteristics between the two treatment groups, respectively. The issue of unmeasured values in some covariates (due reasonably to a “missing at random” (MAR) mechanism [19]) was handled by using the multiple imputation method, and final estimates of the coefficients and standard errors were obtained by pooling model results on ten imputed datasets [20]. After the evaluation of baseline characteristics, all the preoperatively significant (*p* < 0.05) variables were then tested for balance and employed as weights in the inverse probability weighting (IPW) analysis. This was conducted to balance the oncologic risk between the two populations. Moreover, IPW was preferred to avoid the best patients’ selection as in the case of propensity score matching. The model was fitted to each of the 10 datasets to estimate the probability of receiving upfront surgery, conditional on possible confounders. For each patient, a weight was calculated as the inverse of the probability of the treatment actually being received. Final weights were obtained by averaging over the imputed datasets. After obtaining two weighted pseudo-populations, survival analyses were made by the Kaplan–Meyer method, and comparisons among the two groups were made by a robust test. To better stress the impact of the treatment on survival even after the IPW, a double robust test by Cox regression analysis has been made.

Thus, a mortality prediction model was built using a survival Random-Forest (RF) approach. RF was preferred to increase the accuracy of the predictions [21]. To create the model, training (70% of cases) and test (30%) samples were generated randomly. The model was tuned via 10-fold cross-validation without replacement. The log-rank was used as the splitting rule, 100 trees were growing, and the survival time to fit the model was fixed at 60 months. The number of variables to possibly split at each node (“mtry”) was the square root of the total number of variables included in the model. The predictive performance was then evaluated using Harrell’s C-index. For each patient, the model coefficients were used to simulate the potential overall survival under each treatment (neoadjuvant or upfront). Subsequently, the best potential treatment (BPT) within patients was determined as the one leading to the highest predicted OS. Once the BPT for each patient was established, a classification and regression tree for machine-learning (CART) model was developed to select the most important variables determining the allocation of each patient to his/her BPT and to develop a decision tree. The algorithm of the decision tree model works by recursive portioning, which means the data were repeatedly partitioned into multiple subspaces to lead the final sub-space outcome as homogenous as possible [22]. With this approach, the algorithm produced a set of rules to predict the overall mortality by repeatedly splitting the predictor variables. The first employed variable was the one with the highest association with the outcome. Stopping criteria to end the splitting were: (1) all leaf nodes are pure with a single class; (2) a pre-specified minimum number of training observations that cannot be assigned to each leaf node with any splitting methods; and (3) the number of observations in the leaf node reaches the pre-specified minimum one [23]. Once the tree has grown, pruning is performed to minimize overfitting by using the lowest complexity parameter (cp), assessed by the one-minus standard error rule (reflecting the trade-off between the complexity of the model and how it fits the data). Cp was established at 0.0108. To assess the predictive performance of the final model, the concordance index (c-index) was calculated with the bootstrapping resample method. At the end of the procedure, a tree was graphically represented: the first split was, as mentioned, the variable with the highest association with overall mortality, while going down through the tree, the variables were reported by the model in order of importance. Thus, in the final leaf, the graph reported the assigned treatment according to the algorithm, together with the probability to be submitted to the neoadjuvant regimen and the probability to be assigned to upfront surgery.

All tests were two-tailed, and the accepted level of significance was 5%. Analyses were made with R open software (4.0.6, libraries: WeightThem, cobalt, RFSRC, mice, rpart, rpart.plot).

## 3. Results

### 3.1. Observed Results before Weighting

Between 2008 and 2020, 509 patients were consequently treated in our center with a first diagnosis of CLM. One-stage liver resection was considered feasible in all cases, and consequently, no conversion chemotherapy was administered. Thirty-two (6.1%) were excluded because they were lost at follow-up. Moreover, 29 (5.7%) patients were excluded because, after neoadjuvant preoperative chemotherapy, a progression of disease (PD) was found, and they were excluded from the surgical program. Finally, 448 patients were enrolled: 95 (21.2%) in the UPS group and 353 (78.8%) in the NEOS one. There were several significant differences between the two groups: median age (*p* < 0.001), N status of the primitive tumor (*p* = 0.001), rate of metachronous tumors that was higher in the UPS group (*p* < 0.001), median number of lesions (*p* < 0.001), rate of concomitant extrahepatic spread of the disease, vascular contact (*p* < 0.001), and rate of planned R1vasc (*p* = 0.001). These and other baseline characteristics, including data regarding chemotherapy, were summarized in Table 1.

The median follow-up was 43 months (IQR 22–66). In terms of survival, the median OS was 60 months (95% CI: 51-NA) and 41 months (95% CI: 34–46) for UPS and NEOS, respectively (*p* = 0.0036). The observed survival curve is represented in Figure 1A.

### 3.2. Comparison after Inverse Probability Weighting

All the preoperative and oncologic variables that were significantly different between the two groups were employed to make an IPW obtain two well-weighted pseudo-populations. After that, the UPS group was composed of 432.17 pseudo-patients and the NEOS of 439.85. The balance after weighting among variables is depicted in Figure 2 (boxplot of the weights) and Appendix A, Table A1 (mean differences among variables before and after the weighting).

After the weighting, NEOS showed a significant increase in the risk of mortality (HR 1.82, 95% CI: 1.05–3.14, *p* = 0.031) at the univariate level (Figure 1B). However, after a double robust test with a multivariate Cox regression, the presence of concomitant extrahepatic spread (HR 1.90, 95% CI: 1.18–2.90, *p* = 0.007), blood transfusion during the recovery (HR 1.80, 95% CI: 1.10–2.90, *p* = 0.019), and being N2 at the primitive staging (HR 3.70, 95% CI: 1.80–7.70, *p* < 0.001) were independently associated with the risk of mortality (Appendix A, Table A2).

### 3.3. Modelling the Risk of Overall Mortality by Random-Forest and Assessing the Best Potential Treatment Per Each Patient

The mortality risk model developed by Random-Forest (100 trees trained) showed a Harrel’s C-Index of 0.66, and the out-of-bag error was 0.34. The variables’ importance and the relative error rates were depicted in Figure 3.

Once the mortality prediction model was developed, it was applied to every single patient, employing their own clinical and oncologic characteristics, but the treatment was externally fixed to simulate the risk of mortality in case all the cohorts were treated with NEOS. Then, the same process was made to simulate the predicted risk in case all patients were treated by UPS. After this process, every single patient had a mortality risk prediction, according to the model, under both treatments: the one providing the longer OS was considered for that patient as the BPT.

By this application of the model, 364 patients were assigned to the BPT-NEOS group and 84 to the BPT-UPS group. Male patients were significantly more frequent in the BPT-UPS group (79.8% vs. 55.5%, *p* < 0.001). The vascular detachment was needed more in the BPT-NEOS group (56.3% vs. 17.9% in the BPT-UPS group, *p* < 0.001). Consequently, R1vasc was planned and achieved more frequently in the BPT-NEOS group (54.4% vs. 10.7%, *p* < 0.001). BPT-NEOS group had a higher preoperative median CEA (7.05, IQR 3.00–31.50 versus 4.25 IQR 2.68–10.40 in the BPT-UPS, *p* = 0.034), and the median number of metastases was 5 (IQR 2–11) and 6 (4–14) for the BPT-NEOS and BPT-UPS groups, respectively (*p* = 0.013). Median tumor size was higher in the BPT-NEOS group (3.50 cm, IQR 2.00–5.00) than in the BPT-UPS groups (2.50 cm, IQR 1.80–4.58, *p* = 0.023). The baseline of the BPT cohort was summarized in Table 2. Comparing the data observed for real and adopting the treatment indication suggested by our model, 85 (23.4%) BPT-NEOS patients were treated by UPS, while 279 (76.6%) were treated with NEOS; conversely, BPT-UPS patients were treated with UPS in 10 (21.9%) cases, while 74 (88.1%) received NEOS.

### 3.4. Classification and Regression Tree (CART)

To better select patients who should be candidates for BPT-NEOS or BPT-UPS according to our model and to create homogeneous groups, a CART tree was developed. The method selected to prune the algorithm considered the following variables, in order of importance in determining the allocation: planned R1vasc, number of intrahepatic metastases, colon tumor localization, CEA, and sex. The decision tree is depicted in Figure 4, reporting the estimated probability to be assigned to both treatments per leaf. After applying the CART model to the test population, the accuracy rate was 88.6%.

## 4. Discussion

Our model suggests that while neoadjuvant perioperative chemotherapy plus surgery is a well-established treatment that is correctly allocated in most cases (77%), upfront surgery is delivered appropriately in less than 22% of cases. A big room for growth is evident in our data to better understand who the best candidates for both treatments would be.

Considering the observed data, patients who were submitted to upfront surgery showed a more favorable disease: they were oligometastatic, often with a synchronous presentation, and had a lower tumor burden. This is not surprising, reflecting a common practice in which neoadjuvant treatments are reserved for the most advanced cases [2]. As already stated, the ESMO guidelines indicate UPS for those with a technically feasible disease and an excellent oncologic prognosis. However, our series demonstrated that under a parenchyma-sparing policy, surgical feasibility was strongly boosted [24], technically enabling a direct liver resection in almost all cases. From this point of view, no need for conversion chemotherapy was observed, thus satisfying one of the two criteria of the ESMO guidelines for receiving UPS. On the other hand, this new approach relies on a new concept in surgical oncology: the R1 vascular. Indeed, a CLM in contact without invasion with the main intrahepatic vessel can be detached with a free recurrence rate and overall survival as per R0 resections [7]. Interestingly, after weighing the oncologic and clinical risk factors, the present data showed no clear advantage in terms of risk of mortality by submitting patients to preoperative chemo rather than UPS, as demonstrated in other series [8] and in the EPOC trial [25]. These results may be due to an inappropriate allocation to UPS rather than NEOS, reflecting the real-life difficulties of making the best-tailored choice in each case. This consideration drove us to create a machine-learning algorithm to better help in allocating patients to the most favorable oncologic strategy according to their characteristics.

By creating a random-forest model with good accuracy in predicting the risk of mortality, we simulated the potential survival under an upfront rather than a neoadjuvant perioperative approach, taking into consideration all the patient-specific tumor characteristics. Thus, our model enables us to estimate the risk of mortality under both treatments, permitting us to understand, in terms of potential treatment, which could be the most favorable tailored approach in the specific oncologic setting. By creating an automatic decision tree, we identified the most relevant variables that influence the choice to allocate a patient to a neoadjuvant regimen rather than an upfront resection. Interestingly, the need for an R1vasc resection has been identified as the most important factor in the decision process. In this setting, perioperative chemotherapy administration seems to be the best choice in any case.

When liver metastases are not in contact with a vascular structure and the number of metastases is more than 4, the best potential treatment is conditioned by the patient and tumor characteristics. A neoadjuvant regimen is the best potential choice in the case of a patient with an unfavorable presentation, as for females or males with primary right colon or transversum cancer [26,27], or in cases of high CEA. Upfront surgery, indeed, can be considered in the case of male patients with a tumor in the left colon and a low CEA. Of note, when the tumor burden is very high (here estimated by the model as a number of nodules ≥ 13), the advantage of perioperative neoadjuvant chemotherapy is lost, and upfront surgery becomes the best potential treatment.

This latter implication should be deeply evaluated. The biology selection operated by the perioperative chemotherapy seems less important in the case of a high number of liver nodules; in fact, chemotherapy increases the risk of wasting time permitting in some cases the disease to progress and pushing out the patient to his or her chance to be cured radically [25,28,29]. In these extreme cases, considering the opportunity of postponing chemotherapy after surgery (adjuvant regimen) should not be considered heretical.

The factors we identified to make the most appropriate allocation are already well-known in their prognostication roles. Primary colon cancer localization is a well-established prognostic factor [26,27]. It has been shown that right-sided colon tumors commonly present with poor prognostic factors such as RAS and BRAF mutations other than microsatellite instability (MSI) [30]. Moreover, they are linked to gender, being more frequent in women, with morbidity and mortality increase in female patients over 65 years old, probably due to loss of estrogen protection that could increase microsatellite instability [31].

The tumor burden, represented by the number of nodules, is still the most important factor that is evaluated: it not only modifies the curative strategy by permitting or not a resection approach, but it has also demonstrated directly related to the aggressiveness and the risk of relapse after a curative approach [3]. In this sense, in the latter years, biological serum markers, such as the CEA, were demonstrated to be a surrogate parameter to estimate the biological malignancy, increasing the ability to stratify patients’ prognosis and probability of recurrence [32].

The present study has several limitations. First, our model permits us to identify the best treatment in terms of oncological risk. However, even if an upfront surgery may be the best choice, its feasibility is highly dependent on various other factors that are not depicted by our model (e.g., hospital facilities, the surgeon’s experience and skills, the adoption of a full US-guided PSR policy, etc.). Moreover, when compared to others [33], the present study seems to lack in analyzing some baseline patients’ characteristics as the genetic mutation profile. This is because of several reasons: on one side, the referral nature of our center led patients arriving at our evaluation as a second or third opinion after initial disease management performed elsewhere. This results in a difficult retrieval of information processes about their mutational status. On the other hand, data were not available in most of the cases due to the time period of this study, which included some years in which the genetic profile was not routinely analyzed in all cases. Furthermore, it is important to state that, at present, artificial intelligence models are not thought to substitute human decisions but to augment our ability to make good predictions. In this sense, our model should be interpreted as a simulator that may help multidisciplinary meetings determine the best potential oncological approach, allowing them to make their own decisions according to their potentialities. A model, in fact, is thought of as a support and not as a replacement for the physicians’ professionalism and experience. Another significant limitation is the retrospective and monocentric nature of this study. Our daily practice is well known to be oriented to boost the limits of parenchyma-sparing surgery, but this policy is still underemployed worldwide. In our cohort, all patients were treated by surgery, before or after chemotherapy; the fact that some well-recognized prognostic factors, such as the extrahepatic spread, did not result as an effect modifier in our model could be explained by the fact that in this cohort the eventual spread has always been resected effectively. Moreover, from a statistical point of view, our model should be externally validated to confirm its ability to make accurate mortality predictions and good patient allocations. This risk was mitigated by a bootstrap resampling internal validation; however, the external one remains mandatory.

## 5. Conclusions

In conclusion, based on an automated machine-learning analysis of our data, the radical liver clean-up in case of CLM by UPS should be considered more frequently than the present, particularly in the more advanced disease. Further studies are needed to confirm or disprove this insight.

## Figures and Tables

**Figure 1 cancers-15-00613-f001:**
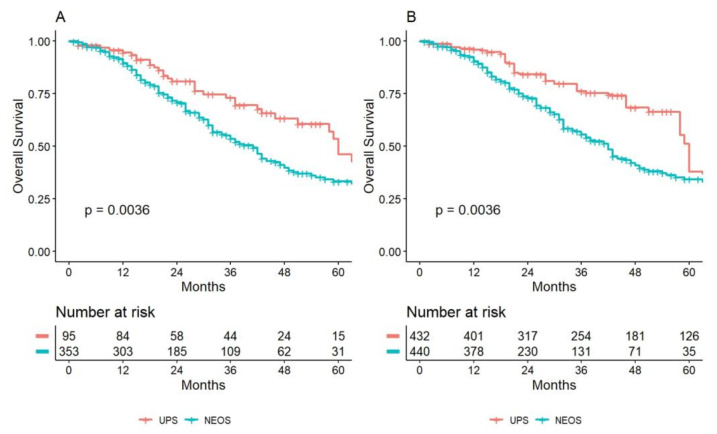
Overall survival is depicted among the two groups (**A**) before and (**B**) after the Inverse Probability Weighting. (UPS Upfront Surgery; NEOS neoadjuvant plus surgery).

**Figure 2 cancers-15-00613-f002:**
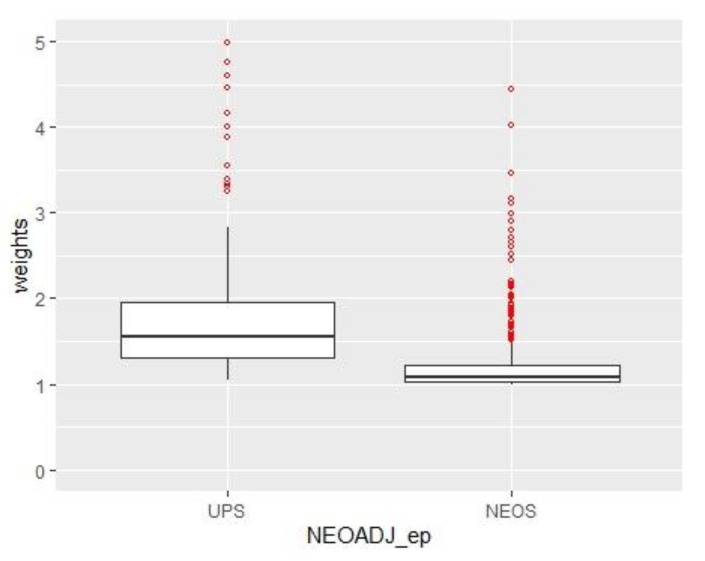
Boxplot with the distribution of the weights among the groups after the weighting.

**Figure 3 cancers-15-00613-f003:**
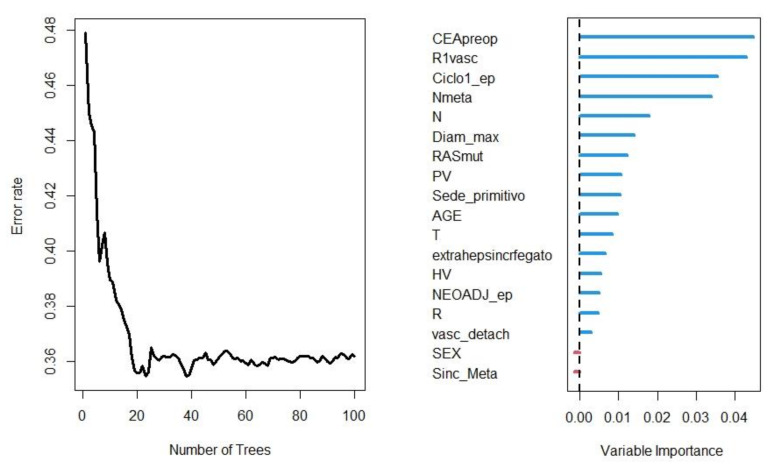
Error rates per each forest and the variables’ importance and the relative coefficient, obtained after the Random-Forest analysis.

**Figure 4 cancers-15-00613-f004:**
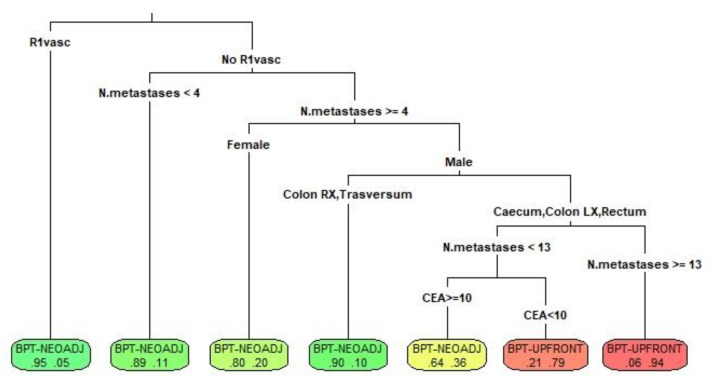
The decision tree obtained by CART is here depicted. The first number in the last leaf represents the probability to be submitted to a neoadjuvant regimen, while the second number is the probability to be assigned to upfront surgery. (Colon RX: right colon tumor, Colon LX: left colon tumor).

**Table 1 cancers-15-00613-t001:** Baseline characteristics of the cohort of patients analyzed.

	UPS	NEOS	*p*
N	95	353	
**Male (%)**	59 (62.1)	210 (59.5)	0.731
**Age, years (median [IQR])**	68.00 [59.00, 74.50]	61.00 [53.00, 68.00]	<0.001
**N colon cancer (%)**			0.001
**N0**	39 (41.1)	82 (23.2)	
**N1**	32 (33.7)	163 (46.2)	
**N2**	16 (16.8)	94 (26.6)	
**NA**	8 (8.4)	14 (4.0)	
**T colon cancer (%)**			0.67
**T1**	5 (5.3)	14 (4.0)	
**T2**	17 (17.9)	64 (18.1)	
**T3**	53 (55.8)	211 (59.8)	
**T4**	13 (13.7)	50 (14.2)	
**NA**	7 (7.4)	14 (4.0)	
**Primary localization (%)**			0.263
** *Caecum* **	2 (2.1)	20 (5.7)	
** *Right Colon* **	23 (24.2)	63 (17.8)	
** *Trasversum* **	6 (6.3)	10 (2.8)	
** *Left Colon* **	41 (43.2)	165 (46.7)	
** *Rectum* **	22 (23.2)	91 (25.8)	
**NA**	1 (1.1)	4 (1.1)	
**RAS MUT (%)**	33 (34.7)	178 (50.4)	<0.001
**NA**	16 (16.8)	10 (2.8)	
**Metachronous disease (%)**	67 (70.5)	69 (19.5)	<0.001
**N° metastases (median [IQR])**	2.00 [1.00, 5.00]	6.00 [4.00, 13.00]	<0.001
**Size max (median [IQR])**	3.70 [2.25, 5.15]	3.10 [2.00, 5.00]	0.211
**CEA preop (median [IQR])**	8.00 [2.50, 35.00]	6.30 [3.00, 27.40]	0.723
**Synchronous extrahepatic (%)**	6 (6.3)	74 (21.0)	0.002
**Number of NEOADJ cycles (median [IQR])**	-	7.00 [5.00, 11.00]	<0.001
**CHT regimen**			-
** *Irinotecan* **	-	94 (26.6)	
** *Oxaliplatin* **	-	224 (63.5)	
** *5FU/Capecitabin* **	-	11 (1.2)	
** *Irinotecan-Oxaliplatin* **	-	24 (6.8)	
**Target therapies**			-
** *Anti-EGFR* **	-	100 (28.3)	
** *Anti-VEGF* **	-	113 (32.0)	
**RECIST (%)**			<0.001
** *Complete Response (CR)* **	-	7 (2.0)	
** *Partial Response (PR)* **	-	283 (80.2)	
** *Stable disease (SD)* **	-	59 (16.7)	
**OSH (%)**	17 (17.9)	154 (43.6)	<0.001
**N resection areas (median [IQR])**	2.00 [1.00, 3.00]	3.00 [2.00, 5.00]	<0.001
**Vascular detachment (%)**	31 (32.6)	189 (53.5)	<0.001
**H-zone (%)**	26 (27.4)	150 (42.5)	0.01
**P-zone (%)**	8 (8.4)	80 (22.7)	0.003
**Blood transfusion (%)**	11 (11.6)	49 (13.9)	0.678
**Length of stay (median [IQR])**	9.00 [8.00, 11.00]	9.00 [8.00, 13.00]	0.396
**R1vasc (%)**	29 (30.5)	178 (50.4)	0.001
**Steatosis (%)**	19 (20.0)	96 (27.2)	0.196
**Liver adjuvant CHT (%)**	34 (35.8)	198 (56.1)	0.001

(NA: not available; RAS MUT: mutated RAS gene; CEA: carcinoembryogenic antigen; NEOADJ: neoadjuvant therapy; CHT: chemotherapy; 5FU: 5 fluoro-uracil; EGFR: epidermal growth factor receptor; VEGF: vascular endothelial growth factor; RECIST: Response Evaluation Criteria in Solid Tumors; OSH: One Stage Hepatectomy; R1vasc: R1 vascular resection).

**Table 2 cancers-15-00613-t002:** Baseline characteristics of the best potential treatment cohorts estimated after the creation of the model for mortality risk by Random-Forest.

	BPT-NEOADJ	BPT-UPFRONT	*p*
N	364	84	
**Male (%)**	202 (55.5)	67 (79.8)	<0.001
**Age, years (median [IQR])**	62.00 [54.00, 70.00]	61.00 [56.75, 69.00]	0.703
**Tumor localization (%)**			<0.001
** *Caecum* **	19 (5.2)	3 (3.6)	
** *Right Colon* **	86 (23.6)	2 (2.4)	
** *Trasversum* **	13 (3.6)	3 (3.6)	
** *Left Colon* **	164 (45.1)	44 (52.4)	
** *Rectum* **	82 (22.5)	32 (38.1)	
**pN (%)**			0.074
**0**	98 (26.9)	33 (39.3)	
**1**	170 (46.7)	31 (36.9)	
**2**	96 (26.4)	20 (23.8)	
**pT (%)**			0.156
**1**	16 (4.4)	4 (4.8)	
**2**	62 (17.0)	23 (27.4)	
**3**	231 (63.5)	48 (57.1)	
**4**	55 (15.1)	9 (10.7)	
**RAS MUT (%)**	186 (51.1)	37 (44.0)	0.296
**Metachronous presentation (%)**	114 (31.3)	22 (26.2)	0.43
**Number of metastases (median [IQR])**	5.00 [2.00, 11.00]	6.00 [4.00, 14.00]	0.013
**Size, cm (median [IQR])**	3.50 [2.00, 5.00]	2.50 [1.80, 4.58]	0.023
**CEA (median [IQR])**	7.05 [3.00, 31.50]	4.25 [2.68, 10.40]	0.034
**Concomitant extrahepatic spread (%)**	58 (15.9)	22 (26.2)	0.04
**Observed-NEOS (%)**	279 (76.6)	74 (88.1)	0.03
**Number of CHT cycles (median [IQR])**	6.00 [4.00, 10.00]	7.00 [4.00, 10.00]	0.029
**Vascular detachment (%)**	205 (56.3)	15 (17.9)	<0.001
**H—Zone (%)**	164 (45.1)	12 (14.3)	<0.001
**P—Zone (%)**	83 (22.8)	5 (6.0)	0.001
**R1vasc (%)**	198 (54.4)	9 (10.7)	<0.001

(RAS MUT: Mutated RAS gene, CEA: carcinoembryogenic antigen, CHT: chemotherapy, R1vasc: R1 vascular resection).

## Data Availability

The retrospective analysis was performed using data from the adult patients enrolled in the liver unit registry and was conducted according to the guidelines of the Declaration of Helsinki. The data sets generated and/or analyzed during the current study are not publicly available but are available from the corresponding author upon reasonable request.

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
