# Peer review of "Upfront Surgery versus Neoadjuvant Perioperative Chemotherapy for Resectable Colorectal Liver Metastases: A Machine-Learning Decision Tree to Identify the Best Potential Candidates under a Parenchyma-Sparing Policy"

_cancers, 2023, doi:10.3390/cancers15030613_

Round 1

Reviewer 1 Report

The manuscript is an innovative study written in a clear way. It offers a new, modern tool to help the clinician in the difficult selection process for neoadjuvant systemic chemotherapy followed by surgery or upfront surgery. The study has some limitations, which are well acknowledged by the authors, such as its monocentric, retrospective nature, and the lack of external validation in order to confirm patients’ allocation and mortality prediction. Otherwise, it is an interesting work, with a potentially high impact for clinical decision-making in the field of oncologic surgery.

There are only minor corrections to make before considering it for publication.

Could the authors reinforce the Material and Methods section providing additional details or by improving the graphical representations of their work (e.g. of the decision tree algorithm) ?

 I encountered minor spelling errors through the manuscript.

Author Response

The manuscript is an innovative study written in a clear way. It offers a new, modern tool to help the clinician in the difficult selection process for neoadjuvant systemic chemotherapy followed by surgery or upfront surgery. The study has some limitations, which are well acknowledged by the authors, such as its monocentric, retrospective nature, and the lack of external validation in order to confirm patients’ allocation and mortality prediction. Otherwise, it is an interesting work, with a potentially high impact for clinical decision-making in the field of oncologic surgery.

There are only minor corrections to make before considering it for publication.

Could the authors reinforce the Material and Methods section providing additional details or by improving the graphical representations of their work (e.g. of the decision tree algorithm)?

Thank you for the suggestion, we have now added more sentences as required.

 I encountered minor spelling errors through the manuscript.

We apologize for the inconvenience. We deeply and carefully checked the text and made modification accordingly.

Reviewer 2 Report

The study investigated best potential treatment (BPT) between neoadjuvant chemotherapy followed by liver resection (NEOS) or upfront surgery (UPS) for colorectal-liver-metastases (CLM) patients. This is a significant problem to solve as current guideline is qualitative rather than quantitative on patient evaluation for treatment option. The authors analyzed 448 patients in a single center through a mortality risk model after weighting baseline differences between two retrospective patient groups. They also constructed a classification-and-regression-tree to quantify and identify candidates for each treatment, which would be very useful for clinicians. They interpreted their machine learning outcome in meaningful clinical context and contributed to the counterintuitive insight that the radical liver clean-up in case of CLM by UPS should be considered more frequently than the present clinical practice. Overall, the manuscript is well written with sound method, meaningful results, and logical discussion. Minor clarifications and language polishing would improve this great work even further.

The authors compared demographic and clinical characteristics to identify and balance oncologic risk factors between the two patient cohorts. How did the authors determine which baseline characteristics of the patients get analyzed? This recent publication seems to include more factors than the authors examined in their study (https://www.frontiersin.org/articles/10.3389/fonc.2022.973418/full)

There are a few statistical options for conditioning on potential confounders, including restriction, matching, adjustment and weighting. Would be helpful if the the authors can provide rationale for using inverse Probability weighting in this study. Along the same lines, it would be helpful if the authors could explain why they chose the specific machine-learning method (Random-Forest) for assessing the best potential treatment.

Use accurate medical terms: Page 2, Line 80 “primitive colorectal cancer” - does the authors mean “primary colorectal cancer” instead?

Page 6, Line 199-200: it’s a bit odd to have patient numbers that are not integers. Please consider editing.

Author Response

The study investigated best potential treatment (BPT) between neoadjuvant chemotherapy followed by liver resection (NEOS) or upfront surgery (UPS) for colorectal-liver-metastases (CLM) patients. This is a significant problem to solve as current guideline is qualitative rather than quantitative on patient evaluation for treatment option. The authors analyzed 448 patients in a single center through a mortality risk model after weighting baseline differences between two retrospective patient groups. They also constructed a classification-and-regression-tree to quantify and identify candidates for each treatment, which would be very useful for clinicians. They interpreted their machine learning outcome in meaningful clinical context and contributed to the counterintuitive insight that the radical liver clean-up in case of CLM by UPS should be considered more frequently than the present clinical practice. Overall, the manuscript is well written with sound method, meaningful results, and logical discussion. Minor clarifications and language polishing would improve this great work even further.

The authors compared demographic and clinical characteristics to identify and balance oncologic risk factors between the two patient cohorts. How did the authors determine which baseline characteristics of the patients get analyzed? This recent publication seems to include more factors than the authors examined in their study (https://www.frontiersin.org/articles/10.3389/fonc.2022.973418/full)

Thank you for the interesting comment. We routinely and prospectively collect all the clinical data available for each single patient treated in our centre. These data come from all the in-hospital diagnosis, assessment, and treatment protocol, and all the variables reported in the paper are the information we collected from each single cases. Since the retrospective nature of the paper, some factors that may affect the results could be not collected, and we stressed this issue in the study limitation section at the end of the discussion. In particular, when our baseline is compared with the one reported in the paper suggested, it seems that our paper lacks the genetic characteristics. This is because of the referral nature of our centre: most of the patients came as second or third opinion after an initial disease management performed in other centres. Moreover, the completed genetic data were not available in most of the cases: this is due to the time-period of the study that included some years in which the genetic data were not routinely collected in all cases (before the spread of the evidence of the impact of such mutations in the patients’ prognosis). Indeed, we added these comments in the discussion and we add the reference suggested.

Considering the weighting, the variables included in the model were selected according to their significance (p<0.05) at the baseline among groups. We better explain this in the methods section.

There are a few statistical options for conditioning on potential confounders, including restriction, matching, adjustment and weighting. Would be helpful if the the authors can provide rationale for using inverse Probability weighting in this study. Along the same lines, it would be helpful if the authors could explain why they chose the specific machine-learning method (Random-Forest) for assessing the best potential treatment.

Thank you for the suggestion, we modify the text accordingly.

Use accurate medical terms: Page 2, Line 80 “primitive colorectal cancer” - does the authors mean “primary colorectal cancer” instead?

We apologize for the error, we modified accordingly.

Page 6, Line 199-200: it’s a bit odd to have patient numbers that are not integers. Please consider editing.

We perfectly understand this confusion. However, when IPW is applied, the model created two “pseudo population”, which means that they are not real patients (we better specify this in the method section). This procedure generates decimals, and for clear methodological reasons we reported the exact number of the generated pseudopopulation. To avoid confusion, we changed the word “patients” with “pseudo patients”. However, in the rest of the results, and in the survival curves, patients count is reported as integer.